

# Framework to prioritize watersheds for diffuse pollution management in Korea: application of multicriteria analysis using the Delphi method

Gyumin Lee[1], Kyung Soo Jun[2], Minji Kang[3]

[1]Construction and Environmental Research Center, Sungkyunkwan University, Suwon, 16419, Republic of Korea
[2]Graduate School of Water Resources, Sungkyunkwan University, Suwon, 16419, Republic of Korea
[3]Water Environment Policy Division, Ministry of Environment, Sejong, 30103, Republic of Korea

*Correspondence to*: Minji Kang(skyjina@korea.kr)

**Abstract.**

This study aimed to develop a risk-based approach for determining control areas to manage non-point source pollution, developing a framework to prioritize catchments by considering the characteristics of polluted runoff from non-point sources. The best management, decision-making, and scientific approaches, such as TOPSIS and Delphi technique, are required for the designation of control areas and the application of the best management practices to the control areas. Multicriteria decision-

making methods can handle the diversity and complexity of non-point source pollution. The Delphi technique was employed for selecting the assessment criteria/sub-criteria and determining their weights. Sub-criteria for each catchment unit were scored with either a quantitative or qualitative scale. All non-point pollution sources in South Korea mainland were included, with the exception of a few islands, with catchment prioritization and pollution vulnerability evaluations shown as thematic maps. This study contributes to the field by developing a new risk-based approach for ranking and prioritizing catchments;

this provides valuable information for the Ministry of Environment to use to identify control areas and manage non-point source pollution.

## 1 Introduction

Diffuse pollution (Non-point source pollution) is a major issue in water quality management and catchment management (Hoppe et al., 2004; Huang and Xia, 2001; Lee and Bae, 2002; Orr et al., 2007). Pollutants accumulated on diverse diffuse

sources generally moves by runoff and makes water quality problems worse. The water quality problems caused from diffuse pollution are influenced by meteorological, hydrologic and demographic characteristics of catchments. Insights and tools addressing complexity and uncertainty of the problems are required to solve the problems.

The diversity and complexity of diffuse pollution can be described by Catchment-based risk assessments (Candela et al. 2009; Wang and Yang, 2008) and the assessments can be undertaken by multi-criteria analysis (Huang et al., 2013). The multi-

criteria analysis is suitable to draw a consultation for management on a complicated system. The approach is practical to deal



with many decision problems in environmental management which involve multiple conflicting evaluation criteria as well as a large number of spatial units (Zhang and Huang, 2011). Diverse methods were applied broad in water-related matter because catchment-based water management is complex and interactive due to the inherent trade-offs between social, political, ecological and economic factors (Kiker et al., 2005). Giupponi et al. (1999) developed a multi-criteria analysis system for

producing risk maps of agricultural pollution. Munafo et al. (2005) developed a potential non-point pollution index (PNPI) to assess the global pressure on surface water bodies. Zhang and Huaug (2011) developed a GIS-based multi-criteria analysis method to assess the potential contributions of different land areas in diffuse nutrient export at the basin scale. Chun et al. (2012) took a risk-based approach to prioritize catchments for diffuse metal pollution management. Huang et al. (2013) employed multi-angle indicators of non-point source pollution, deficient waste treatment, and public awareness of

environmental risk to identify key environmental risk sources contributing to water eutrophication and to suggest certain risk management strategies for rural area.

The Ministry of Environment (MOE) of Korea has tried to deal with diffuse pollution issue after the 2000s. The MOE currently enforces programs for the control of diffuse pollution under the Water Quality and Ecosystem Conservation Act (WQECA): the report on facility installation to reduce diffuse pollutants from new development sites or industrial sites, the designation

and management of control areas to be required to manage diffuse pollution, and so on. The programs have been implemented under insufficient data, tools, information and knowledge for diffuse pollution. It makes it difficult to assess existing diffuse pollution, to establish the measures including selection and spatial allocation of management practices, or to evaluate the measures. Recently, sound decision making system and efficient resource allocation are significant issues in the Korean diffuse pollution management. The current decision making support system should be reorganized based on expert advice and

scientific basis for more efficient policy implement.

We were interested in the development of a decision-aiding tool for the selection/designation of control areas. This study aimed to develop a framework to evaluate and to prioritize Korean watersheds in terms of the need of diffuse source management. The above-mentioned multi-criteria analysis was used for the purpose. Moreover, the Delphi method to obtain the most reliable consensus of a group of experts (Dalkey and Helmer, 1963; Linstone and Turoff, 1975, 2002; Okoli and Pawlowski, 2004)

was employed to reflect expert's opinions in the framework development.

## 2 Methodology

### 2.1 Background and study procedure

The Ministry of Environment in South Korea has a plan to continuously expand diffuse pollution management areas until 2020. In 2009, the candidate management areas for industrial sites and small watersheds had been additionally determined for



conducting a feasibility study and preparing selection criteria. Relevant authorities also established the Second Comprehensive Plan for Diffuse Pollution Management (´12~´20) in a collaborative project. As a part of this plan, an improvement scheme(draft) for the criteria of determining and assessing management areas has been prepared and implemented in order to expand diffuse pollution management areas and improve the related systems. Accordingly, the order of priority needs to be set

on a scientific basis so that more desperate areas for diffuse pollution management could be systematically preferred. For the diffuse pollution policy, a vulnerability analysis and a map of vulnerable areas are required.

Some studies (Bang et al., 1997; Yoon et al., 2007; Choi et al., 2009; Park et al., 2014) by the Ministry of Environment and the Korea Environment Corporation have been conducted so far on the nomination of diffuse pollution management area. However, most of these studies assumed scenarios after considering significant factors and merely analyzed the scenarios to

predict the results of assumptions. Consequently, determining, allocating and scoring the evaluation items and weights have become key research issues to identify vulnerable areas to non-point source pollution. To identify and prioritize vulnerable areas for diffuse pollution management, the evaluation items to be made as well as the weighting and scoring methods for each item are to be importantly considered and determined. In this regard, many experts have constantly expressed the opinion that research and survey are needed to propose specific indexes quantifying the contribution rates and weights, and to determine a

selection method for the vulnerability assessment (Park et al., 2013).

The degradation of water quality and aquatic ecosystem due to diffuse pollution sources is related to uncertain factors such as emission characteristics of various pollutants, and climate and soil properties. Naturally, diverse solutions are being proposed by experts and interested parties, which prevent the clear policies from being firmly implemented. For efficient policy implementation, a quantitative, objective and scientific analysis needs to be conducted for factors causing diffuse pollution,

and then the management areas are to be expanded. This will result in not only systematic policy implementation but also highly efficient low investment.

The process of the study to prioritize watersheds for diffuse pollution management in Korea is presented in Fig. 1.

Step 1 sets an evaluation framework and performs a Delphi survey for experts to determine evaluation items and weights.

Step 2 collects and quantifies data of each evaluation item for each watershed. The TOPSIS method, which is one of MCDM

techniques, is applied for evaluation.

Step 3 prepares the vulnerable area map by using evaluation results, and selects major vulnerable areas.

## 2.2 Determining criteria with modified Delphi method

The Delphi method was developed by the RAND Corporation in the 1950's aimed to reduce the range of group responses and to strive for expert consensus. The Delphi method as a method for structuring a group communication process is accomplished

by some feedback of individual contributions of information and knowledge, some assessment of the group judgment or view, some opportunity for individuals to revise views and some degree of anonymity in individual responses (Linstone and Turoff, 1975). A series of questionnaires with controlled opinion feedback is typically used for collecting and distilling knowledge from a group of experts (Rowe et al., 1991; Adler and Ziglio, 1996; Angus et al., 2003).The process that experts reply to





questionnaires, subsequently receive feedback, and modify their opinion is repeated until arriving at the most reliable consensus.

The Delphi method is effective in allowing a group of individuals, as a whole, to deal with a complex problem. (Mohorjy and Aburizaiza, 1997) and has been applied in various fields such as information system, planning, environmental impact
assessment, social policy and public health (Angus et al., 2003; Linstone and Turoff, 2002; Okoli and Pawlowski, 2004). Also, there were several applications in studies on water resources uses and management, water quality assessment and so on (Cude, 2001; Kim et al., 2013; Lee et al., 2013; Parparove et al., 2006; Parparov and Hambright, 2007).

For successful progress, there are two important considerations. First important thing is to select experts participating in our survey. We considered that the respondents should be experts with plenty of experiences and have a high level of responsibility.
Second important thing is to reach an agreement involving experts. Making consensus will take a long time as far as that goes because every experts have their own opinion that sometimes it is extremely different.

We have simplified Delphi process. The nub of modified Delphi procedure is that experts were provided detailed and concrete information for candidate criterion by organizing group. Fig. 2 explain the procedure of our modified Delphi.

Firstly, we selected a group of experts in diffuse pollution management of Korea. The experts are ones who have extensive
experiences with a high level of responsibility or have carried out a lot of research on diffuse pollution. Then we created an evaluation framework with the candidates of criteria and sub-criteria after examining literature and brainstorming. This is because it is time-consuming and inefficient to reach an agreement after every expert suggests an evaluation framework. A questionnaire was prepared to obtain opinion of experts for the framework's structure, the candidates of criteria and sub-criteria. After collecting and analyzing the judgments of a group of experts, the evaluation framework consisting of criteria and their
weights was determined if the consensus of the group emerged.

## 2.3 Assessing potential risk with TOPSIS

In this study, the criteria scores were estimated by the Technique for Order of Preference by Similarity to Ideal Solution (TOPSIS) method of multi-criteria decision analysis methods (Fishburn, 1967; Hwang and Yoon, 1981). The TOPSIS choose the alternative of the shortest geometric distance from the positive ideal solution and the longest geometric distance from the
negative ideal solution (Lai et al., 1994; Chu, 2002; Jun et al., 2011; Lee et al., 2013).

In addition, assessment results for all alternatives can be easily calculated and presented from multi- attribute perspective (Kim et al., 1997; Shih et al., 2007; Lee and Chung, 2007; Chung and Lee, 2009).

Here the Positive Ideal Solution (PIS) is the most vulnerable area and the Negative Ideal Solution (NIS) is the lowest vulnerable area. The TOPSIS procedure is as follows:
Construct the weighted decisions matrix ($v_{ij}$))

$$v_{ij} = w_j \times x_{ij} \tag{1}$$




Where $w_j$ is the weight of j$^{th}$ criterion , $x_{ij}$ are built by alternative $A_j$ $(j = 1, \cdots, n)$ which are evaluated against criteria $C_i$ $(i = 1, \cdots, m)$. the standardized data of each assessment unit area.

Determine the PIS($A^+$) and NIS($A^-$) of unit area

$$A^+ = v_1^+, \cdots, v_n^+ \tag{2a}$$

$$A^- = v_1^-, \cdots, v_n^- \tag{2b}$$

Here $v_i^+ = \max(v_{ij}), \ v_i^- = \min(v_{ij})$

Calculate the distance from the positive ideal ($d_i^+$) and the negative ideal ($d_i^-$) solution for each alternative

$$d_i^+ = \left\{ \sum_{j=1}^n (v_{ij} - v_j^+)^2 \right\}^{1/2}, \ i = 1 \cdots, n \tag{3a}$$

$$d_i^- = \left\{ \sum_{j=1}^n (v_{ij} - v_j^-)^2 \right\}^{1/2}, \ i = 1 \cdots, n \tag{3b}$$

Calculate the optimum membership degree ($D_i^+$)

$$D^+ = \frac{d_i^-}{d_i^+ - d_i^-} \ , \ (i = 1, \cdots, m) \tag{4}$$

The priority of watersheds in terms of the need of diffuse pollution management was decided according to the criteria scores aggregated for watersheds. The priority of diffuse pollution management was represented in a map with a geographic information system (GIS).

## 3 Application

### 3.1 Study Area

The MOE shall regularly survey the kinds of sources of pollution in order to ascertain the current status of water quality and aquatic ecosystems by river-system spheres of influence and shall develop the basic plan for preserving the water quality and aquatic ecosystems (WQECA Article 22 and Article 23, 1997). The spatial extent of the investigation is the whole country of Korea.

The river-system spheres of influence are classified by small, medium and large areas of influence. In the study, the potential risk of diffuse pollution was evaluated for 814 watersheds which are the small areas of influence and the subjects of the pollution survey.

### 3.2 Determination of the evaluation framework by the Delphi survey

The evaluation framework, criteria and their weight were determined by the expert's agreement through the Delphi survey. The weights of multi-criteria were decided by the ranking method.

A nonpartisan expert pool and asked them whether they will participate or not. Total 12 experts gave us a positive answer. They worked in the following different sectors: government-funded research institutes (8%), private engineering companies





(17%), public servant (17%) and university (58%). All had doctoral-level training and most (80%) have been researching on diffuse source pollution management over 10 years.

Candidates of criteria and sub-criteria were organized based on brainstorming with literature research. Then, the draft questionnaire including candidate criteria was distributed for experts. This is the start of the first round. After we collected and

analyzed the raw data from the questionnaires, we then revised the questionnaire. The modified questionnaire included the analysis result of previous survey. The round processed in the same way continued until consensus emerges.

A draft framework to prioritize the watersheds by evaluating the potential risk of diffuse pollution was developed in consideration of the availability of data related to diffuse pollution in Korea and the characteristics of diffuse pollution discussed in other studies (Chon et al., 2012; Novotny, 2002; Jang et al., 2012; Jung et al., 2011; Park et al. 2010). Although

the diffuse pollution is irregular, variable and indefinable and its risk varies depending on the watershed (Candela et al., 2009; EA, 2007; U.S.EPA, 1997), the 'source-pathway-receptor' concept is applicable and useful for the evaluation. Chon et al. (2012) defined criteria of activities and land-use representing pollution source, rainfall and runoff characteristics, physical, chemical, and ecological status of receptor adopting the concept. Similarly, Jang et al. (2012) used the characteristics related to the process of generation, discharge and delivery to receiving waters of agricultural areas. In this study, pollution source,

hydrologic process, and status of receiving water were employed as the groups to classify criteria.

The criteria of land use, activities in urban areas and agricultural areas and the sub-criteria such as population density, livestock numbers, fertilizer use, and area of different land use were selected for the pollution source; the criteria of rainfall and runoff and the sub-criteria such as annual rainfall, rainy days, drainage area, and runoff ratio for the hydrologic process, the criteria of water resource, water quality and aquatic ecosystem and the sub-criteria such as river flow, water quality of BOD, SS, and

the indicator of aquatic ecosystem health for the status of receiving water. The criteria and the sub-criteria of the draft of evaluation framework are as shown in Table 1.

The Delphi survey was carried out to get the approval of experts for the evaluation framework. We selected the total 13 experts with intention of participating in the survey. The experts with experience, education and training at the doctoral-level have been working for government-funded research institutes, government-affiliated organizations or universities and most of them

have been involved in the diffuse source pollution management over 10 years. The experts were asked to check the structure of the evaluation framework, to exclude/add the criteria and the sub-criteria and to decide their weight. At the first round, they supported the group and the criteria of the draft of evaluation framework but requested to modify some of the sub-criteria. The sub-criteria were classified by 'Acceptance', 'Reject', and 'Addition' and were revised for the second round. The feedback on the modified questionnaires was positive and the expert's consensus was built as shown in Table 2.

In this study, two type of the weight sets selected by the expert panels are used. First type is the ranking sets ($w_i^{rank}$). For a given set of variables, each participant determines the ranking in the order of its highest value for each of the factors in a manner that determines and quantifies its importance. Thus, the most important factor becomes rank 1, and the next most important one is rank 2. At this point, the component may be ranked in the same order. The constructed ranks are aggregated


through the conversion, with Rank 1 being converted into m-1 and Rank 2 being converted into m-2. Where m is the total number of factors. Calculate the order of these conversions by the following eq. (5) and (6).

$$R_i = \sum_{j=1}^{n} R_{ij} \tag{5}$$

$$w_i^{rank} = R_i / \sum_{i=1}^{m} R_i \tag{6}$$

where $R_i$ is the sum of transformed ranks, $R_{ij}$ is the transformed rank what the rank selected by the j$^{th}$ panel of experts subtracted from the total number of criteria is, m is the total number of criteria, n is the total number of expert panels.

The weights of the group, the criteria and the sub-criteria were determined by the ranks suggested by the experts. The experts judged that the pollution source (0.4853) is more important than the hydrologic process or the receiving water in the Group. Second type of weights set is the rating sets ($w_i^{rate}$). It is a way to compare importance of criteria to distribute weights. The

survey respondents will determine the weights for each criteria, but will then select values within the range given. The range of weights is a continuous section, generally ranging from 0.0 to 1.0 or up to 100.0. In addition, the sum of the weights given to all variables under comparison is equal to the maximum range given. A factor equivalent to 0.0, the lowest limit of the section, is of no importance to the assessment, whereas a maximum value means that a maximum number of possible values of significance are applied.

The weights may be calculated from the following eq. (7) and (8). It is also possible to set the same value by allocating the relative importance of each factor.

$$w_{ij} = p_{ij} / \sum_{i=1}^{m} p_{ij} \tag{7}$$

$$w_i^{rate} = \sum_{j=1}^{n} w_{ij} / \sum_{j=1}^{n} \sum_{i=1}^{m} w_{ij} \tag{8}$$

$p_{ij}$ is the weight of criteria i, determined by Panel j.

The evaluation framework thus established can be applied flexibly in various conditions including securing of relevant data. In other words, if data are insufficient or uncertain, evaluations are conducted either by removing or applying such insufficiency or uncertainty, and the evaluation results are analyzed to improve the framework. This "Adaptive Management" method is an iterative approach (Holling, 1978; Waters, 1986) that enhances the management ability by accumulating accurate understanding and knowledge of response for a target system.

**3.3 Collection, quantification and standardization of evaluation data**

(1) Data collection and quantification for each evaluation item

Data such as population density, urbanization level and fertilizer use were collected from the source of either the statistics of local governments or the Statistics Korea, and the precipitation data including annual rainfall were provided by the Korea Meteorological Administration. In addition, land use map, industrial condition, combined/sanitary sewer system, livestock

numbers, livestock barn area, watershed map, water resource and water quality were derived from the data surveyed by the Ministry of Environment (Table 4).





When values of each evaluation item for 814 small watersheds were determined, some data items were not measured or missed. As for the water quality data, if there is a water quality observatory in a watershed, the data were obtained from it, and if there is no such observatory and thus no measurement was available, data from an adjacent watershed or lake were analyzed and utilized. As for flow rate, if a small watershed consists of a single basin, the measurement of flow rate is attributable to the

watershed. On the other hand, if there is another upstream small watershed, the measurement cannot indicate the characteristic of a single small watershed. In order to improve this problem, the flow rate, rainfall, and areas of the upstream basin and small watershed were used to calculate a specific discharge and determine the flow rate of each small watershed.

$$Q_2 = \frac{P_2}{P_1}\frac{A_2}{A_1}Q_2 \qquad\qquad (9)$$

where, Q is flow rate, P is rainfall and A is the basin area at the calculation point of flow rate. The subscript 1 means the

reference point, and the subscript 2 indicates the calculation point of flow rate.

Since many small watersheds including estuaries do not have any measurement even at the level of middle watersheds, such watersheds were left unmeasured and a low score was given without using data of adjacent small watersheds. In addition, if necessary, flow rate data of a dam were also utilized to represent runoff characteristics of small watersheds.

(2) Standardization of evaluation items

Because each dataset for evaluation items has different units and properties, standardization is required to use datasets for evaluation. The re-scaling method was adopted in the standardization process. The overall range of data was normalized to assign values between 0 and 1, as described in the equation (10).

$$X_i = \frac{x_i - x_{min}}{x_{max} - x_{min}} \qquad\qquad (10)$$

where, $X_i$ is the i-th standardized value, $x_i$ is the i-th data value, $x_{max}$ is the maximum value, and $x_{min}$ is the minimum value. However, in case the data collected are used to standardize evaluation items without modification, the standardized scores are often either biased or equalized in their range and distribution according to characteristics and types of data. Accordingly, since it seemed to be unreasonable to apply the above equation with no modification, the data collected were prioritized and the consequential order of priority was scored before the equation was used for standardization.

**3.4 Assessing Vulnerability**

The vulnerability of every small watershed to diffuse pollution was evaluated by using data and weights for each factor, and the vulnerable areas were determined based on this assessment (Fig. 3). In addition, the small watersheds were prioritized again in each of 4 large watersheds, and top 30 small watersheds are illustrated in Fig. 4. This was because the pollution source management and relevant policies were organized based on the large watersheds. Both ranking and ratio methods were applied

to calculate weights.

Among top 50 small watersheds in the order of priority in each large watershed, main rivers and small watersheds, which required diffuse pollution source management, were derived in each river system.



Han River basin has 3 priority control target river: down stream of Namhan River, Mid-down stream of Han River and Anseong stream. Geum River basin has 4 priority control target river: mid stream of Geum River, Dongjin River, Mankyeng River and Sapgyo stream.

Youngsan River basin has 2 priority control target river: Youngsan River and Sumjin River. Most vulnerable area of Nakdong

River basin is in main stream of Nakdong River.

The evaluation results were analyzed in terms of effects of each evaluation factor. It turned out that if a large number of livestock are reared and much fertilizer is used in a basin, the land area is wide and the public water has much soil and high SS concentration, such a watershed needs to be preferentially managed.

**3 Conclusion**

There are little studies to assess watersheds in respect of the diffuse pollution management in Korea. This study has suggested a scientific analysis method for selecting priority areas in the current diffuse pollution management system. As various uncertain factors are included in assessing vulnerable areas to diffuse pollution sources, such factors need to be quantified and analyzed objectively and scientifically. The Delphi method was used to determine the vulnerability evaluation items, which

included basin characteristics, pollution source and water quality, and weights for diffuse pollution, on the basis of expert opinions. Criteria and sub-criteria were allocated into three groups of pollution source, hydrologic process, and receiving water. Based on the weights and evaluation items thus obtained, data of each item were applied, and the vulnerability to diffuse pollution was assessed by the TOPSIS method. The proposed evaluation process will promote efficient policy implementation and set a foundation for scientific/clear diffuse pollution management.

In addition, this study attempted a small watershed-based analysis for more selective/intensive policy enforcement. However, it was difficult to standardize quantitatively each evaluation item, which was needed to determine management areas, at the level of small watershed. Accordingly, a runoff model needs to be applied to improve the estimations for unmeasured areas. A vulnerability assessment system for diffuse pollution is also to be established in order to promote efficient policy enforcement. Such system should update relevant data and enable cyclic reevaluation.

Finally, this study has not reflected the current diffuse pollution management policy in the list of evaluation items. It was because the effect of the policy could not be accurately quantified. A further study will solve this problem and include the current policy in assessment.



**Acknowledgement**

This study was financially supported by the Basic Science Research Program of the National Research Foundation of Korea (NRF), which is funded by the Ministry of Education (NRF-2016R1A6A3A11932509).

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


**1** Conducting Delphi survey of experts

- Determining an evaluation framework
- Determining criteria and their weights

**2** Assessing potential risk for diffuse pollution

- Assessing each criterion for watershed units
- Aggregating the scores of all criteria
  for watershed units

**3** Prioritizing watersheds

- Ranking watersheds to need diffuse pollution
  management
- Making a potential risk map
  for diffuse pollution

Figure 1: Study procedure.





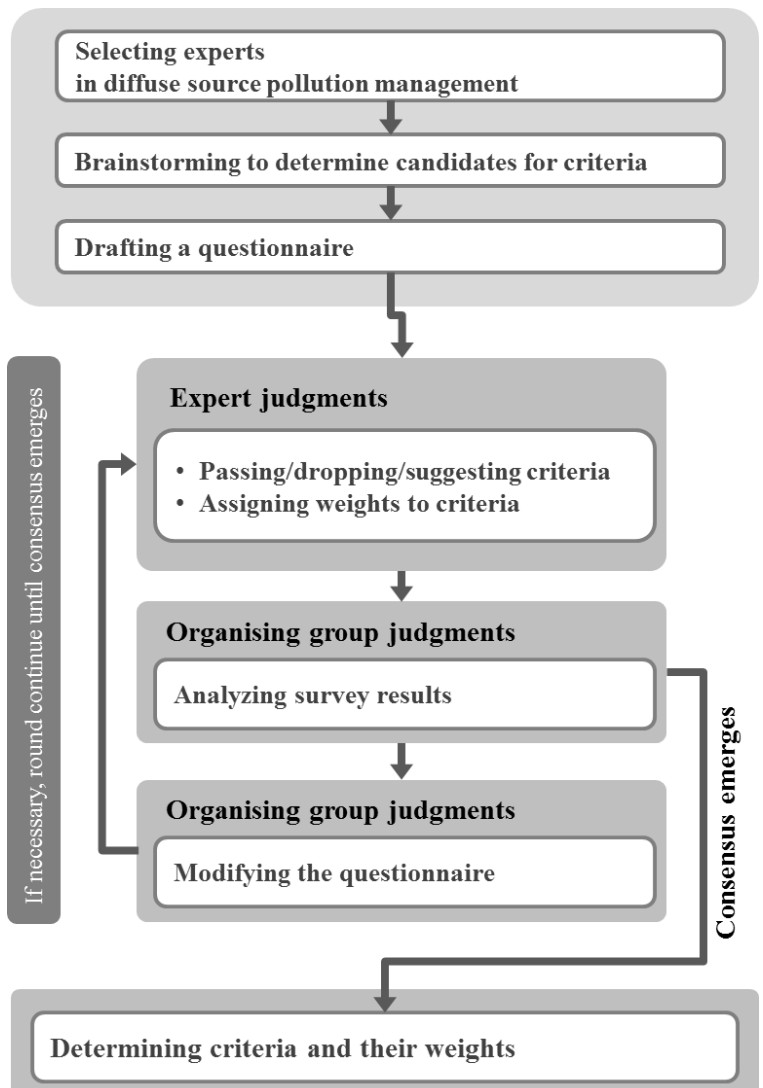

Figure 2: Delphi survey procedure.



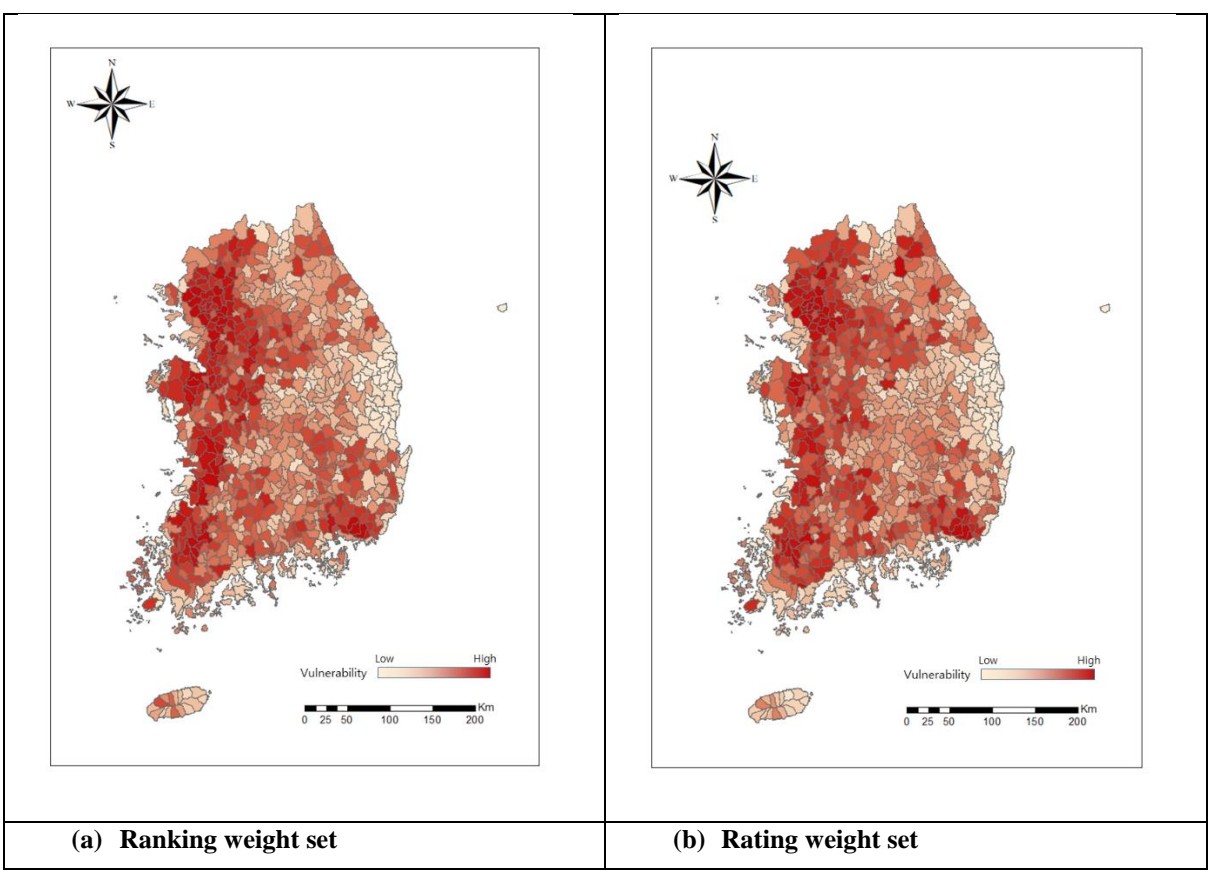

| (a) Ranking weight set | (b) Rating weight set |

Figure 3: Spatial diffuse pollution vulnerability results for South Korea.


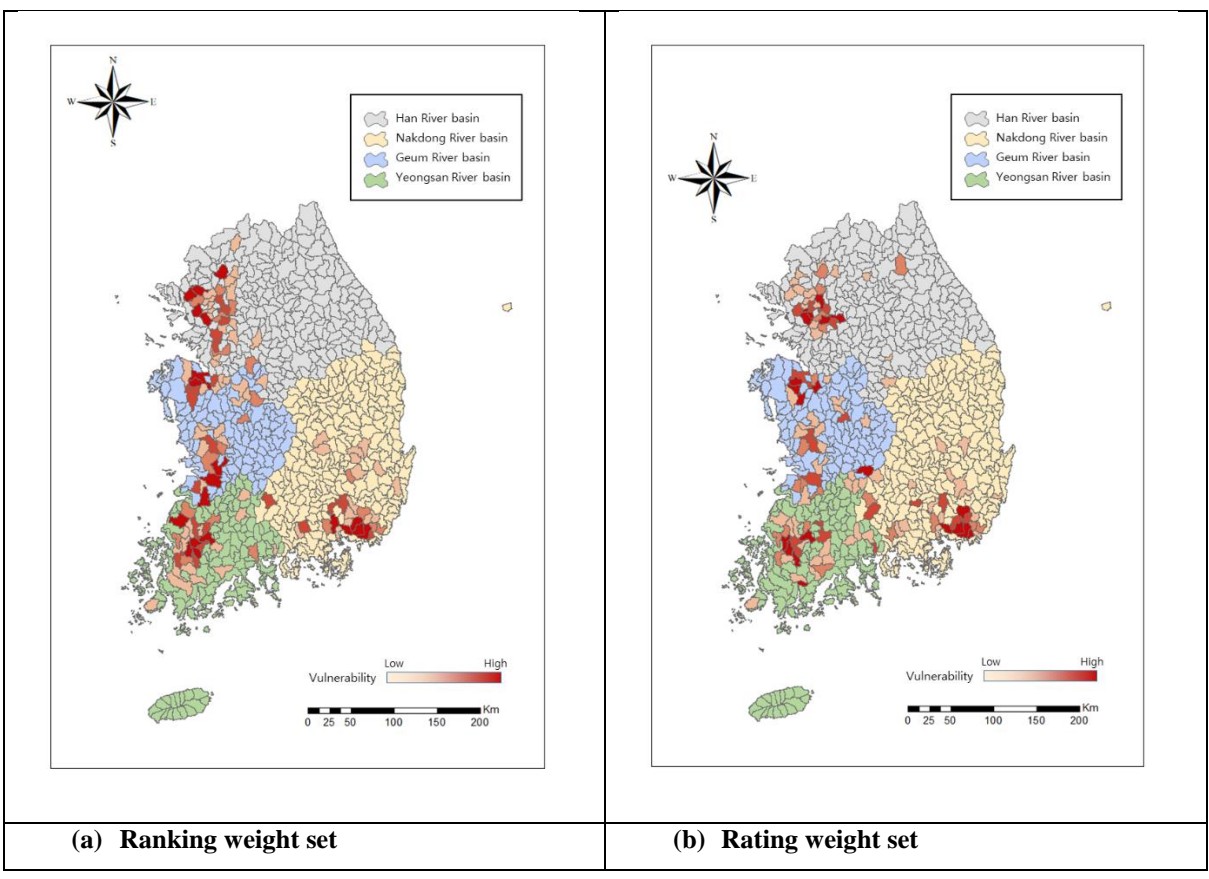

| (a) Ranking weight set | (b) Rating weight set |
|---|---|

Figure 4: Top 30 small watersheds of diffuse pollution vulnerability for each 4 large watersheds.





**Table 1. A set of criteria and sub-criteria for the draft of evaluation framework**

| Group | Pollution source | | | Hydrologic process | | Receiving water | | |
|---|---|---|---|---|---|---|---|---|
| Criterion | Activities in urban areas | Activities in agricultural areas | Land use | Rainfall | Runoff | Water resource | Water quality | Aquatic ecosystem |
| Sub-criterion | · Population density<br>· Urbanization level<br>· Industrial condition | · Total area of Fish farm<br>· Livestock numbers<br>· Livestock barn area<br>· Fertilizer use | · Ratio of impervious area<br>· Road area<br>· Farming area<br>· Forest area | · Annual rainfall<br>· Rainy days<br>· Maximum rainfall | · Drainage area<br>· Runoff ratio<br>· Soil permeability | · River flow<br>· River improvement | · General items (BOD, TN, T-P)<br>·Items for muddy water (SS, turbidity)<br>· Other items | · Aquatic ecosystem health |





**Table 2. Determination of the evaluation framework by the Delphi survey**

| Group | Criterion | 1st round Sub-criterion | Judgment | 2nd round Sub-criterion | Judgment |
|---|---|---|---|---|---|
| Pollution source | activities in urban areas | Population density | Acceptance | Population density | Acceptance |
| | | Urbanization level | Acceptance | Urbanization level | Acceptance |
| | | Industrial condition | Acceptance | Industrial condition | Acceptance |
| | | - | Addition | Combined/Sanitary sewer system | Acceptance |
| | activities in agricultural areas | Total area of Fish farm | Reject | - | - |
| | | Livestock numbers | Acceptance | Livestock numbers | Acceptance |
| | | Livestock barn area | Acceptance | Livestock barn area | Acceptance |
| | | fertilizer use | Acceptance | fertilizer use | Acceptance |
| | Land use | Ratio of impervious area | Modification | urban area(impervious area)* | Acceptance |
| | | Road area | Modification | | Acceptance |
| | | Farming area | Modification | Paddy area* | Acceptance |
| | | | Modification | Farming area* | Acceptance |
| | | Forest area | Modification | Forest area* | Acceptance |
| | | - | - | Other areas* | Acceptance |
| Hydrologic process | Rainfall | Annual rainfall | Acceptance | Annual rainfall | Acceptance |
| | | Rainy days | Acceptance | Rainy days | Acceptance |
| | | Maximum rainfall | Reject | - | - |
| | | - | Addition | Average rainfall intensity | Acceptance |
| | | - | Addition | Average rainfall duration | Reject |
| | Runoff | Watershed area | Acceptance | watershed area | Acceptance |
| | | Runoff ratio | Modification | Land cover | Modification (CN) |
| | | Soil permeability | Acceptance | Soil permeability | |
| | | - | Addition | Watershed Shape | Acceptance |
| | | - | Addition | Average slope of a watershed | Acceptance |
| Receiving water | Water resource | River flow | Acceptance | River flow | Acceptance |
| | | River improvement | Acceptance | River improvement | Acceptance |
| | Water quality | BOD, TN, T-P | Acceptance | BOD, TN, T-P | Acceptance |
| | | SS, turbidity | Acceptance | SS, turbidity | Modification (SS) |
| | | Other items | Acceptance | Other items | Acceptance |
| | Aquatic ecosystem | Aquatic ecosystem health | Acceptance | Aquatic ecosystem health | Acceptance |



**Table 3. Determination of the weights by the Delphi survey**

| Group | Weights | | Criterion | Weights | | Sub-criterion | Weights | |
|---|---|---|---|---|---|---|---|---|
| | $w_{Rank}$ | $w_{Rate}$ | | $w_{Rank}$ | $w_{Rate}$ | | $w_{Rank}$ | $w_{Rate}$ |
| Pollution source | 0.4853 | 0.5077 | Activities in urban area | 0.3333 | 0.3300 | Population density | 0.0479 | 0.2597 |
| | | | | | | Urbanization level | 0.0336 | 0.2338 |
| | | | | | | Industrial condition | 0.0427 | 0.2687 |
| | | | | | | Combined/Sanitary sewer system | 0.0375 | 0.2378 |
| | | | Activities in agricultural areas | 0.3611 | 0.3500 | Livestock numbers | 0.0696 | 0.3850 |
| | | | | | | Livestock barn area | 0.0361 | 0.2350 |
| | | | | | | Fertilizer use | 0.0696 | 0.3800 |
| | | | Land use | 0.3056 | 0.3200 | Urban area | 0.0452 | 0.3380 |
| | | | | | | Paddy area | 0.0349 | 0.2200 |
| | | | | | | Farming area | 0.0392 | 0.2400 |
| | | | | | | Forest area | 0.0179 | 0.1060 |
| | | | | | | Other areas | 0.0111 | 0.0960 |
| Hydrologic process | 0.2206 | 0.2218 | Rainfall | 0.3714 | 0.3960 | Annual rainfall | 0.0361 | 0.3767 |
| | | | | | | Rainy days | 0.0229 | 0.2867 |
| | | | | | | Average rainfall intensity | 0.0229 | 0.3366 |
| | | | Runoff | 0.6286 | 0.6040 | Watershed area | 0.0319 | 0.2269 |
| | | | | | | Curve Number | 0.0540 | 0.3594 |
| | | | | | | Watershed shape | 0.0196 | 0.1718 |
| | | | | | | Average slope of a watershed | 0.0331 | 0.2419 |
| Receiving water | 0.2941 | 0.2705 | Water resource | 0.2985 | 0.3000 | River flow | 0.0585 | 0.7200 |
| | | | | | | River improvement | 0.0293 | 0.2800 |
| | | | Water quality | 0.4925 | 0.4850 | BOD, TN, T-P | 0.0476 | 0.3000 |
| | | | | | | SS | 0.0692 | 0.4900 |
| | | | | | | Other items | 0.0281 | 0.2100 |
| | | | Aquatic ecosystem | 0.2090 | 0.2150 | Aquatic ecosystem health | 0.0615 | 1.0000 |



**Table 4. Methods of collection and quantification for Criteria**

| Group | Criterion | Sub-criterion | Methods of collection and quantification |
|---|---|---|---|
| Pollution source | Activities in urban area | Population density | Population per unit area (man/km²) |
| | | Urbanization level | Percentage of the total population living in urban areas |
| | | Industrial condition | Estimation with the business scale[1], **the emission of** specific/specified substance harmful to the quality of water[2], and the tolerance area of the quality standard[3] 1) level 1 – 5 point, level 2 – 4 point, level 3 – 3 point, level 4 – 2 point, level 5 – 1 point 2) emission – 2 point, non–1 point 3) Exceptional area 4 points, clean area 3 points, area type 1– 2 points, area type 2– 1 point |
| | | Combined/Sanitary sewer system | Estimate the sum of values with the weights[1] using the ratio of the sewage system, combined sewer system, the area without public sewage system 1) the ratio of the sewage system 0.25, the ratio of combined sewer system 0.5, the ratio of the area without public sewage system 1.0 |
| | Activities in agricultural areas | Livestock numbers | Estimation of livestock numbers with the weights1) by the type of livestock[1] 1) The type of livestock on Total Maximum Daily Loads system parts 2) Wastewater unit Discharge Flow and Loading Rates by the type of livestock (Milk cow 0.5673, Korean native cattle 0.1816, Horse 0.1207, Pig 0.1070, Sheep and deer 0.0087, Dog 0.0137, Poultry 0.0010) |
| | | Livestock barn area | Estimate the sum of farm area on the type of livestock |
| | | Fertilizer use | Total consumption of nitrogen, phosphorous, and fertilizer per year |
| | Land use | Urban area | Calculate the sum of urban area in land use |
| | | Paddy area | Calculate the sum of paddy area in land use |
| | | Farming area | Calculate the sum of farming area in land use |
| | | Forest area | Calculate the sum of forest area in land use |
| | | Other areas | Calculate the sum of other area in land use |
| Hydrologic process | Rainfall | Annual rainfall | Calculate the sum of rainfall per year |
| | | Rainy days | Calculate the sum of rainy days per year |
| | | Average rainfall intensity | Mean value of hourly rainfall intensity of the rainfall event per year |
| | Runoff | Watershed area | Area of watershed |
| | | Curve Number | Calculate the average CN Number using land cover area and hydrologic soil group |
| | | Watershed shape | Shape Factor[1] 1) Basin shape factor is the ratio of basin length to effective basin width |
| | | Average slope of a watershed | Calculating mean slope aspect each cell in DEM (Digital Elevation Model) |
| Receiving water | Water resource | River flow | Average discharge of measuring points |





| | River improvement | Assessment score of inhabitation/waterside environment [1]<br>1) natural longitudinal and transverse shoal of river, width of Riverside, sediment quality, river-crossing structures, channel characteristics, embankment material, land use in inner and outer land, treatment Facilities, and etc. |
|---|---|---|
| Water quality | BOD, TN, T-P | Calculate the sum of standardized[1] BOD, TN, TP on measuring points<br>1) the ratio of each water watershed to total watershed of average resource |
| | SS | Standardized SS of water quality measurement site |
| | Other items (COD, Chlorophyll a, water temperature) | Calculate the sum of standardized COD, Chlorophyll a, water temperature on measuring points |
| Aquatic ecosystem | Aquatic ecosystem health | Total score of aquatic ecosystem health[1]<br>1) Mean of Trophic Diatom Index of attached algae, Korea Purity biotic Index (KPI) using benthic macroinvertebrates, and Index of Biological Integrity(IBI) of fish |