# Peer review of "Framework to prioritize watersheds for diffuse pollution management in Korea: application of multicriteria analysis using the Delphi method"

_Natural Hazards and Earth System Sciences, 2019_

## Referee Comment (RC1) · Jose Manuel Antón (Referee) · 4 Jun 2019

Dear Sirs, "The paper subject is interesting, as for important actual watersheds water pollution management situations for all South Korea, and corresponds to the NHESS journal. As evident local errors I felt that in page 9 "3 Conclusion" must be "4 Conclusion". Also in page 8 the equation (9), "$\eth\text{ÍŚĎ}2 = (\eth\text{ÍŚČ}2/\eth\text{ÍŚČ}1).(\eth\text{ÌŘť}2/\eth\text{ÌŘť}1).\eth\text{ÍŚĎ}2$ ", is incoherent by including twice the $\eth\text{ÍŚĎ}2$, probably by error.

The paper is about selecting diffuse pollution watersheds that have effects on water of rivers, and the questions are indicated with an extensive bibliography, and that is intended to aid the programs of Ministry of Environment for 2009-2020 years to reduce

diffuse pollutants effects. The paper presents shortly the risk based approach and as referee I can follow it only imprecisely. In 2.1 the procedure is envisaged as indicated in Figure 1, indicating that a modified Delphi method was used. In 2.3 the TOPSIS technique is indicated with its formulae, saying it could be used to assess potential risks. The use of Delphi survey in a special way is indicated in 3.2 with Figure 2. With indication for data in 3.3, in 3.4 it is indicated that general results were obtained, with two examples; from the block of results general trends were indicated in Figures 3 and 4, as maps for all South Korea, that include some 814 watersheds as indicated. In 4 Conclusion, page 9, the study of the paper is considered, indicating global results. They do not get "the current diffuse management policy" that could be attained for a further study. The consideration of the general problem in a resumed text, the extensive and long bibliography, and the very short indication of results seems adequate to indicate somehow the main lines to evaluate policies for improving the quality of water in rivers of Korea. The indicated methods and results could have been more intensively shown, to afford they use. But the paper does not inform on the water qualities considered, on the experts consulted using Delphi method, and on what was done with TOPSIS technique, and it does not precise the results of the studies and their possible use. Some more information on these achieved operations could be a sensible improvement."

Best regards, J. Antón

---

## Referee Comment (RC2) · Jose Manuel Antón (Referee) · 15 Jun 2019

"Thank you for your answer. You have done corrections for the two evident local errors and you have included "Figure 3: Sample of collected data sets." with 6 indication maps of South Korea. The text has been conserved as it was. That is well. You conserve the level of indication for the content that is interesting, connecting with a lot of bibliography, for these important considerations used for all South Korea. As referee I think that the paper is valid for that journal, the subjects are interesting and the information is valid and could have been more precise. "

[Figure]

2019-152, 2019.

---

## Referee Comment (RC3) · Anonymous Referee #2 · 21 Jul 2019

From my point of view, this work represents a good contribution to the research in the field of water quality management. Authors propose an interesting approach for ranking and prioritizing areas that need diffuse pollution management. In this study, multicriteria analysis of non-point source pollution problem is solved by applying TOPSIS and Delphi technique, so that scientific analysis and expert advice can be efficiently involved in the decision making support system. However, it seems that appropriate revisions to the following points should be undertaken: 1. Your results lack DISCUSSIONS. There is no sufficient explanations about final results while many factors (e.g. factors of hydrological process) are considered in the multi-criteria analysis. 2. Page 2, Line 23. "The above-mentioned multi-criteria analysis..." What does it mean? 3.

[Figure]

Page 5, Line 1-2. Please make the description more clearly. What do these variables represent with reference to the specific problem considered in the study? 4. Page 7, Line 5. Is Rij transformed rank? 5. Page 8, Line 5-7. I cannot understand the meaning of this sentence. 6. What does "modification" mean that is mentioned in page 8, line 21. 7. Page 9, Line 21-22. Please rewrite this sentence to make its meaning more clearly. 8. Please correct minor errors. Page 2, Line 2. . . .applied broadly. . . Page 2, Line 6. Zhang and Huang (2011). . . Page 4, Line 13. Fig. 2 explains. . . Page 4, Line 15. . . .a lot of researches. . . Page 4, Line 23. The TOPSIS chooses. . . Page 5, Line 10. First space should be removed. Page 5, Line 27. Please check the grammar of this sentence. Page 6, Line 30. In this study, two types of. . . Page 7, Line 5. "Rij is the transformed rank. . ." Please refine this sentence. The authors are also suggested to check the spelling and grammar throughout the manuscript.

---

## Author Comment (AC3) · 9 Sep 2019

I apologize for the delay in responding to your comments. We are very grateful for your valuable comments. Our research is expected to help the governmental policy in non-point source pollution management. we want to thank you once again for your review.

---

## Author Response (AR1)

Dear Sirs, "The paper subject is interesting, as for important actual watersheds water pollution management situations for all South Korea, and corresponds to the NHESS journal. As evident local errors I felt that in page 9 "3 Conclusion" must be "4 Conclusion". Also in page 8 the equation (9), "ðÍŚĎ2 = (ðÍŚČ2/ðÍŚČ1).(ðÍŘť2/ðÍŘť1).ðÍŚĎ2 ", is incoherent by including twice the ðÍŚĎ2, probably by error.

The paper is about selecting diffuse pollution watersheds that have effects on water of rivers, and the questions are indicated with an extensive bibliography, and that is intended to aid the programs of Ministry of Environment for 2009-2020 years to reduce

diffuse pollutants effects. The paper presents shortly the risk based approach and as referee I can follow it only imprecisely. In 2.1 the procedure is envisaged as indicated in Figure 1, indicating that a modified Delphi method was used. In 2.3 the TOPSIS technique is indicated with its formulae, saying it could be used to assess potential risks. The use of Delphi survey in a special way is indicated in 3.2 with Figure 2. With indication for data in 3.3, in 3.4 it is indicated that general results were obtained, with two examples; from the block of results general trends were indicated in Figures 3 and 4, as maps for all South Korea, that include some 814 watersheds as indicated. In 4 Conclusion, page 9, the study of the paper is considered, indicating global results. They do not get "the current diffuse management policy" that could be attained for a further study. The consideration of the general problem in a resumed text, the extensive and long bibliography, and the very short indication of results seems adequate to indicate somehow the main lines to evaluate policies for improving the quality of water in rivers of Korea. The indicated methods and results could have been more intensively shown, to afford they use. But the paper does not inform on the water qualities considered, on the experts consulted using Delphi method, and on what was done with TOPSIS technique, and it does not precise the results of the studies and their possible use. Some more information on these achieved operations could be a sensible improvement."

Best regards, J. Antón

———————————————————

Nat. Hazards Earth Syst. Sci. Discuss.,
https://doi.org/10.5194/nhess-2019-152-AC1, 2019

[Figure]

We thank for your valuable and very scrupulous comments. I tried to sincerely response to all of your comments as explained below. I have corrected the title number and equation 9 that you have pointed out in page 9 and 8, respectively. I was embarrassed to that mistake. In order to help readers understand, I have included few figures to present the collected data. Since this study used 26 estimation criteria for 814 subbasins that covering the whole country, showing details about data used for estimation and computation procedures in the paper is difficult. In our future study, we will develop

a policy for the management of nonpoint pollution source after selecting priority regions for management. Monitoring will be required for this future study. The purpose of this study is to provide scientific information for the selection of regions where a policy of nonpoint source can be applied. Even though many efforts have been carried out by Department of Environment in Korea for reducing the emissions of pollution source, there are still difficulties in management of industries such as livestock industry that are the main pollution source due to their close relationship with individual income and regional economy. So far, therefore, local governments prioritize regions where technical assistance for the management of nonpoint source is requested. However, this method can not control the regions that really need management. Therefore, this study is trying to provide procedures to achieve water quality goal based on characteristics of regions by monitoring the application of policy scenarios for improvement of water quality in the example areas selected from priority regions. Thank you very much for your consideration.

Best regards, Gyumin Lee

Please also note the supplement to this comment:
https://www.nat-hazards-earth-syst-sci-discuss.net/nhess-2019-152/nhess-2019-152-AC1-supplement.pdf
* * *
[Figure]

[revised manuscript text omitted]

Nat. Hazards Earth Syst. Sci. Discuss.,
https://doi.org/10.5194/nhess-2019-152-RC3, 2019

[Figure]

**NHESSD**
From my point of view, this work represents a good contribution to the research in the field of water quality management. Authors propose an interesting approach for ranking and prioritizing areas that need diffuse pollution management. In this study, multi-criteria analysis of non-point source pollution problem is solved by applying TOPSIS and Delphi technique, so that scientific analysis and expert advice can be efficiently involved in the decision making support system. However, it seems that appropriate revisions to the following points should be undertaken: 1. Your results lack DISCUSSIONS. There is no sufficient explanations about final results while many factors (e.g. factors of hydrological process) are considered in the multi-criteria analysis. 2. Page 2, Line 23. "The above-mentioned multi-criteria analysis..." What does it mean? 3.

[Figure]

Page 5, Line 1-2. Please make the description more clearly. What do these variables represent with reference to the specific problem considered in the study? 4. Page 7, Line 5. Is Rij transformed rank? 5. Page 8, Line 5-7. I cannot understand the meaning of this sentence. 6. What does "modification" mean that is mentioned in page 8, line 21. 7. Page 9, Line 21-22. Please rewrite this sentence to make its meaning more clearly. 8. Please correct minor errors. Page 2, Line 2. . . .applied broadly. . . Page 2, Line 6. Zhang and Huang (2011). . . Page 4, Line 13. Fig. 2 explains. . . Page 4, Line 15. . . .a lot of researches. . . Page 4, Line 23. The TOPSIS chooses. . . Page 5, Line 10. First space should be removed. Page 5, Line 27. Please check the grammar of this sentence. Page 6, Line 30. In this study, two types of. . . Page 7, Line 5. "Rij is the transformed rank. . ." Please refine this sentence. The authors are also suggested to check the spelling and grammar throughout the manuscript.

———————————————————

[Figure]

Nat. Hazards Earth Syst. Sci. Discuss.,
https://doi.org/10.5194/nhess-2019-152-AC2, 2019

[Figure]
Everything you have pointed out has been corrected on attached supplement file. I've marked the corrections in red.

1. Your results lack DISCUSSIONS. There is no sufficient explanations about final results while many factors (e.g. factors of hydrological process) are considered in the multi-criteria analysis. Answer) we improved the conclusion. 2. Page 2, Line 23. "The above-mentioned multi-criteria analysis. . ." What does it mean? Answer) this sentence

was deleted because it's redundant. 3. Page 5, Line 1-2. Please make the description more clearly. What do these variables represent with reference to the specific problem considered in the study? Answer) It is corrected. 4. Page 7, Line 5. Is Rij transformed rank? 5. Page 8, Line 5-7. I cannot understand the meaning of this sentence. Answer) It is corrected. 6. What does "modification" mean that is mentioned in page 8, line 21. Answer) It is corrected. 7. Page 9, Line 21-22. Please rewrite this sentence to make its meaning more clearly. Answer) It is corrected. 8. Please correct minor errors. Answer) It is corrected.

we want to thank you once again for your review.

Please also note the supplement to this comment:
https://www.nat-hazards-earth-syst-sci-discuss.net/nhess-2019-152/nhess-2019-152-AC2-supplement.pdf

———————————————————————

[revised manuscript text omitted]